# The Effect of Swimming on Anxiety-Like Behaviors and Corticosterone in Stressed and Unstressed Rats

**DOI:** 10.3390/ijerph17186675

**Published:** 2020-09-14

**Authors:** Mohammad Amin Safari, Maryam Koushkie Jahromi, Rasoul Rezaei, Hadi Aligholi, Serge Brand

**Affiliations:** 1Department of Sport Sciences, School of Education and Psychology, Shiraz University, Shiraz 71946-84334, Iran; aminsafari70@yahoo.com (M.A.S.); koushkie53@yahoo.com (M.K.J.); rasoulrezai1364@gmail.com (R.R.); 2Department of Neuroscience, School of Advanced Medical Sciences and Technologies, Shiraz University of Medical Sciences, Shiraz 71348-14336, Iran; hadialigholi@yahoo.com; 3Division of Sport and Psychosocial Health, Department of Sport, Exercise and Health, University of Basel, 4052 Basel, Switzerland; 4Center for Affective, Stress and Sleep Disorders, Psychiatric Clinics, University of Basel, 4002 Basel, Switzerland; 5Substance Abuse Prevention Research Center, Health Institute, Kermanshah University of Medical Sciences (KUMS), Kermanshah 6719851115, Iran; 6Sleep Disorders Research Center, Kermanshah University of Medical Sciences (KUMS), Kermanshah 6719851115, Iran; 7Department, School of Medicine, Tehran University of Medical Sciences (TUMS), Tehran 1416753955, Iran

**Keywords:** corticosterone, depression, anxiety, swimming, recovery, stress

## Abstract

This study assessed the effect of swimming training on anxiety-like behaviors and corticosterone. Thirty adult male Wistar rats were randomly assigned to five study conditions: swimming training (ST); exposure to chronic mild stress (CS); exposure to chronic mild stress followed by swimming training (CS + ST); exposure to chronic mild stress followed by a recovery period (CS + recovery); control. The exercise training consisted of 60 min of swimming exercise per day, for five days a week, and four consecutive weeks. A chronic mild stress program (CMS) was applied for a period of four weeks. Anxiety-like behaviors were measured by open field test (OFT). The number of excrements and blood corticosterone were used as physiological parameters of anxiety. To assess corticosterone, blood samples were taken 48 h after the last session of experiments. Compared to other study conditions, the lowest anxiety-like behaviors and corticosterone concentrations were observed in the ST condition in unstressed rats. In stressed rats, as in the ST + CS group, swimming training probably reduced some anxiety behaviors, but the results showed increased corticosterone compared to control and CS + Recovery. Anxiety parameters and corticosterone concentrations were greatest in the CS condition. In the ST group, anxiety parameters were less than for the ST + CS group. In the CS + Recovery group, anxiety parameters were less than for the CS group. In summary, self-paced swimming training could attenuate some anxiety parameters in both stressed and non-stressed rats. The effect of swimming training in unstressed rats was more prominent than in stressed rats. In stressed rats, a period of recovery was more effective than swimming training in reducing corticosterone. Mechanisms of anxiety reduction other than cortisol should be investigated in future research.

## 1. Introduction 

Chronic stress is considered one of the major mental health issues [1]. Not surprisingly, exposure to chronic stress can lead to a broad variety of adverse physiological and psychological effects. More specifically, exposure to chronic stress negatively affects brain functioning [2]. According to evidence from both human and animal studies, exposure to chronic stress is causally related to psychiatric disorders such as major depressive disorders and anxiety disorders [3,4,5,6]. Psychiatric disorders are associated with an altered hypothalamic–pituitary–adrenocortical (HPA) axis activity [7]. Chronic and severe psychological stress leads to the HPA hyperactivity and corticosteroid hypersecretion [8]. Various pharmacological and non-pharmacological methods (e.g., physical activity) are recommended for treatment of anxiety and depression.

Studies in rodents showed that regular physical activity decreases cortisol concentrations and regulates and normalizes noradrenergic and serotonergic concentrations [9,10,11,12,13,14,15]. Exercise can adjust the release of corticotrophin-releasing hormone from the hypothalamus and the adrenocorticotrophic hormone from the anterior pituitary [16], which moderates the responsiveness to stress and anxiety [17]. In rodent studies, different types of exercise were used as a preventive and treatment agent for anxiety [18]. In rats, Wen et al. [19] found that chronic stress caused an overactivation of the mitochondria in the raphe nuclei; by contrast, running training suppressed the effect of overactivation. Furthermore, a number of studies have shown that running training improved mitochondrial function and increased antioxidant enzymes in the nervous tissue. For example, in rats, Dos Santos et al. [20] showed that running training during growth may protect the brain from oxidative damage caused by exposure to chronic stress that occurs later in life. Likewise, Gomes da Silva [21] showed that that mild-to-moderate physical activity increased hippocampal anti-inflammatory cytokine levels in older rats. Moreover, swimming could reduce depressive-like behaviors in rats during pregnancy [22]; swimming increased the activity of the antioxidant enzyme superoxide dismutase (SOD) [23] and reduced oxidative damage by increasing proinflammatory cytokines [22] and SOD activity [24]. Further, and always among animal studies, swimming decreased IL-10 levels and oxidative stress and was associated with improved inflammation and inflammatory mediators (such as TNF-a, IL-6, and CRP) [25]. Finally, in humans, when compared to running, swimming was more effective in reducing inflammatory markers (e.g., IL-6 and cortisol) [26].

Collectively, studies in both humans and non-humans showed the association between symptoms of anxiety and higher cortisol and inflammatory markers [27,28,29]; further, and again in studies with humans and non-humans, swimming turned out to favorably impact symptoms of anxiety, as well as cortisol concentrations and inflammatory markers [23,24,26]. However, to our knowledge, no animal studies have investigated so far whether the favorable influence of swimming training (ST) could be observed also after the exposure to chronic mild stress and compared to a recovery period without further intervention. The aim of the present study was, therefore, to evaluate among the stressed and unstressed rats the effect of swimming on anxiety-like behaviors and physiological responses which was measured through open field indices, corticosterone and number of excrements. 

Given the increasing prevalence of chronic symptoms of stress, the findings of the study could be useful to clarify the effect of swimming as a treatment measure for anxiety or a recovery time for those who have experienced a period of stress.

## 2. Materials and Methods

### 2.1. Animals

Thirty male Wistar rats (age: 10–12 weeks, weight: 220 ± 20 g) were provided from the animal lab at Shahid Beheshti University of Medical Sciences (Shahid Beheshti University Center for Animal Experiment, Tehran, Tehran Province, Iran). The animals were kept in a room with a controlled (temperature 22–25 °C, humidity 40–50%, 12:12 h light–dark cycle) and completely quiet environment without any noise pollution and stressors, and had free access to water (electrical conductivity of 0.64 ms/cm at temperature of 25 °C) and food (consisting of a total of 16.6 kJ/g: carbohydrate 66.40%, fat 10.60% and protein 23%). All animal programs and conservations were provided by the researcher who had a formal degree and experience in this regard. For preventing the possible effect of rats with anxiety on healthy rats, they were housed in separated cages and places. Additionally, all health principles were respected by the researcher in order to prevent possible contamination and animal disease. The laboratory and training environment were washed and cleaned daily [30]. All study procedures on the animals were in accordance with the National Institutes of Health guide for the care and use of laboratory animals (NIH Publications No. 8023, revised 1978) [31]. The ethical and graduate study committee at Shiraz University registered the study proposal with the code number of 941bbc6a-c3b1-4f5e-bfe1-70e8862a28f4.

The rats were divided into 5 groups of 6 rats, including: animals that did not perform any exercise and did not receive any stress (Con), animals that were exposed to chronic stress followed by performing swimming training (CS + ST), animals that were exposed to chronic stress (CS), animals that performed swimming training (ST), animals that were exposed to chronic stress followed by a period of recovery time with neither training nor exposure to stress (CS + R). The treatment can be summarized with following steps. 

Con: no treatment for 10 weeks;CS + ST: (1) 4 weeks of stress, (2) 2 weeks of adaptation with water, (3) 4 weeks of swimming;ST: (1) 2 weeks of adaptation with water, (2) 4 weeks of swimming;CS: (1) 4 weeks of stress;CS + R: (1) 4 weeks of stress, (2) 6 weeks of no treatment.

48 h after the treatments, the OFT was performed and after that the blood samples were taken for corticosterone measurement. Open field testing was performed on all animals 48 h after the experiments. Blood samples were taken 12 h following taking the open field test. A summary of the study protocol is in Figure 1. 

### 2.2. Swimming Training Protocol

Swimming training groups performed 60 min of swimming per day, for 5 days a week, and 4 weeks [32]. All training sessions were performed under red light (as it causes the least amount of stress) at 6 pm, which is the best training time in the mice’s normal activity rhythm [33].

To alleviate stress without promoting adaptation to exercise, all rats were adapted to water prior to initiating the experiment. This adaptation consisted of two parts. The first part focused on water adaptation, which lasted 4 days. On the first day of adaptation, the rats were placed in a shallow water swimming pool for 5 min so that the rats could only stand. On the second day of adaptation, the rats were placed in the pool for 5 min while the pool water was above the animal’s head and the rats could start swimming. On the third day of adaptation, the water was so deep that the rats had to swim for 5 min. On the last day of adaptation, the rats had to swim for 15 min. The water temperature was maintained at 31 °C at all stages [34]. 

The second part of the protocol focused on the rats’ adaptation to the swimming training. It consisted of 6 days and began with 15 min of swimming training. The second day of adapting to the training protocol consisted of 24 min of swimming training. The adaptation was a 10-day process in which rats were adapted to water, the depth of which was increased over the next 4 days so that it was necessary for the rats to swim (first for 5 min, then for 15 min). Over the next 6 days, the duration of swimming increased to 60 min. Thereafter, the rats swam for 60 min per day, 5 days a week for 4 weeks [32] (Table 1).

In order to control the possible effect of environment, the control group of rats were immersed in shallow water at 31 °C for 30 min 5 days per week, for 4 weeks [35].

### 2.3. Chronic Mild Stress (CMS) Protocol

Rats were exposed to mild chronic stress for 21 days, which included 2 h of paired caging, 18 h of free access to food followed by 1.5 h of restricted access to food (0.2 g pellet), 18 h of water deprivation followed by 1.5 h of empty bottle exposure, 21 h of wet cage (300 mL of water added per 100 g of bedding), 36 h of continuous lighting and 3 h of 45° cage tilting. This program of stress was applied for four weeks [28] (Table 2).

### 2.4. Open Field Test

The Open field test was used to evaluate the level of anxiety through locomotor activities [36]. The apparatus consisted of a wooden box measuring 72 × 72 × 36 cm. The floor of the arena was divided into sixteen equal squares: four in the center and twelve in the periphery (four corner and eight wall) [37] (Figure 2). The total distance (the total distance traveled) and average speed (average movement speed) were measured to evaluate general locomotor activity. Additionally, the center time (duration of time spent in a central square), wall time (duration of time spent in a wall square), corner time (duration of time spent in a corner square) and number of excrements were measured to evaluate anxiety. The test duration was 5 min [37,38]. All measurements (times and distances) were recorded by an AHD camera 2-MP, Model No: cp-B20M3, Lens: 3.6 mm-3MP (Shenzhen MVTEAM Technology Co, Ltd, Shenzhen city, China). The analysis was performed in a blind manner. The apparatus was cleaned with a solution of 10% ethanol between trials to eliminate animal clues. 

### 2.5. Blood Sampling and Corticosterone Measurement

After training and taking the open field test, and following an overnight fast, the rats were anesthetized using ketamine/xylazine (10 mg/kg xzlazine + 75–100 mg/kg ketamine), blood samples were collected into a syringe via cardiac puncture. The blood samples were taken into Eppendorf tubes immediately. All samples were collected between 08:00 a.m. and 10:00 a.m. to avoid temporal influences on blood metabolites. After centrifugation at 4000 rpm at 4 °C for 10 min (using Centrifuge Centric 322A, Domel Company, Slovenia), serum samples were separated and stored at −80 °C prior to assay. Plasma corticosterone concentrations were determined by ELISA Kit by detection of 25 ng/mL (ZellBio GmbH, Ulm, Germany). Analysis was conducted according to the manufacturer’s instructions. 20 µL of the sample was poured into 200 µl conjugated enzyme and incubated and maintained for 60 min. Values were calculated according to a standard curve generated during the experiment [18,39].

### 2.6. Statistical Analyses

Shapiro–Wilk tests were used to determine the distribution of normality. Leven’s tests were performed to ensure the homogeneity of variances. One-way ANOVAs were performed to compare the study groups. The independent factor was Group (Con; CS + ST; CS; ST, CS + R); dependent factors were total distance; average speed; center time; corner time; wall time; number of excrements; corticosterone concentrations. For paired group comparisons, post-hoc analyses were performed with the least significant difference (LSD) test. The level of significance was set at *p* < 0.05. All statistical computations were performed with SPSS® 23.0 (IBM Corporation, Armonk, NY, USA) for Windows®.

## 3. Results

Shapiro–Wilk tests showed normal distribution for all dependent variables. Leven’s tests showed homogeneity of variances for all groups. 

Considering the weight of the animals, the comparison of the weight before and after the treatment indicated that in the CON group (230.33 ± 25.28 vs. 233.67 ± 24.44) the weight did not change significantly (*p* = 0.341); in the CS + ST group (229.50 ± 17.76 vs. 232.03 ± 26.67) the weight (229.50 ± 17.76 vs. 215.95 ± 12.42) reduced following the stress (*p* = 0.003), while it did not change significantly following the exercise (p=0.060); in the ST group (256.58 ± 13.54 vs. 224.17 ± 7.46) the weight reduced significantly (*p* = 0.001), and in the ST group (284.36 ± 2.32 vs. 279.83 ± 2.31) the weight did not change significantly (*p* = 0.342).

### 3.1. Total Distance

Total distance was significantly different between the five group conditions: (F (4, 25) = 78.77, *p* < 0.001) (Table 3). Post-hoc analyses showed that compared to the CON (3244.60 ± 272.94), the total distance was significantly shorter in the CS (2179.11 ± 119.23; *p* < 0.001), in the CS + Recovery condition (CS + Recovery (2499.64 ± 200.91; *p* < 0.001) (Figure 3 and Figure 4).

Compared to the CON condition, the total distance was significantly longer in the ST group (4376.49 ± 359.27; *p* < 0.001).

Compared to the CS + ST (3090.42 ± 117.51; *p* = 0.263), total distance was not significantly different in the CON condition.

To summarize, as regards the total distance as a measure of anxiety-like behavior, it was highest in the chronic mild stress condition, compared to the ST, CON, CS + Recovery and CS + ST conditions. By contrast, anxiety-like behavior was lowest in the ST condition, compared to the CS, CON, CS + Recovery, CS + ST conditions.

### 3.2. Average Speed

Average speed was significantly different between the five group conditions: (F (4, 25) = 187.06; *p* < 0.001) (Table 3). Post-hoc analyses showed that compared to the CON (11.41 ± 0.72), average speed was significantly shorter in the CS (4.96 ± 0.61), *p* < 0.001) and in the CS + Recovery conditions (7.18 ± 0.27; *p* < 0.001) (Figure 4 and Figure 5).

Compared to the CON condition, the average speed was significantly longer in the ST condition (15.96 ± 0.83; *p* < 0.001).

Average speed was not significantly different in the CS + ST condition (10.59 ± 1.08; *p* = 0.075) compared to the CON condition.

To summarize, regarding the average speed as a measure of anxiety-like behavior, anxiety was greatest in the chronic mild stress condition, compared to the ST, CON, CS + Recovery and CS + ST conditions. By contrast, anxiety-like behavior was lowest in the ST condition, compared to the CS, CON, CS + Recovery and CS + ST conditions.

### 3.3. Center Time

Center time was significantly different between the five group conditions: (F (4, 25) = 193.96, *p* < 0.001) (Table 3). Post-hoc analyses showed that compared to the CON (24.45 ± 0.35), center time was significantly shorter in the CS (8.42 ± 4.13; *p* < 0.001) and in the CS + Recovery conditions (11.15 ± 0.16; *p* < 0.001) (Figure 4 and Figure 6).

Compared to the CON condition, the center time was significantly longer in the ST (38.45 ± 1.91; *p* < 0.001).

There was no significant difference regarding center time in the CS + ST (22.31 ± 1.12; *p* = 0.095) condition compared to the CON condition.

To summarize, regarding the center time as a measure of anxiety-like behavior, anxiety was highest in the chronic mild stress condition, compared to the ST, CON, CS + Recovery and CS + ST conditions. By contrast, anxiety-like behavior was lowest in the ST condition, compared to the CS, CON, CS + Recovery and CS + ST conditions.

### 3.4. Corner Time

Corner time was significantly different between the five group conditions: (F (4, 25) = 125.82, *p* < 0.001) (Table 3). Post-hoc analyses showed that compared to the CON (202.36 ± 5.17), corner time was significantly longer in the CS (274.40 ± 10.56; *p* < 0.001) condition, in the CS + Recovery condition (243.03 ± 2.99; *p* < 0.001), in the CS + ST condition (225.28 ± 3.30; *p* < 0.001) (Figure 4 and Figure 7).

The corner time was significantly shorter in the ST condition compared to the CON condition (156.70 ± 17.56; *p* < 0.001).

To summarize, as regards the corner time as a proxy for anxiety-like behavior, anxiety was highest in the chronic mild stress condition, compared to the ST, CON, CS + Recovery and CS + ST conditions. By contrast, anxiety-like behavior was lowest in the ST condition, compared to the CS, CON, CS + Recovery and CS + ST conditions.

### 3.5. Wall Time

Wall time was significantly different between the five group conditions: (F (4, 25) = 69.85, *p* < 0.001) (Table 3). Post-hoc analyses showed that compared to the control condition (CON) (61.73 ± 4.89), wall time was statistically significantly shorter in the chronic mild stress condition (CS (13.73 ± 11.11; *p* < 0.001), in the CS + Recovery condition (38.62 ± 3.13; *p* < 0.001) (Figure 4 and Figure 8). Compared to the CON condition, the wall time was significantly longer in the ST (104.85 ± 17.73; *p* < 0.001). Wall time was not significantly different in the CS + ST condition compared to the CS + Recovery condition (40.13 ± 5.43; *p* = 0.796).

To summarize, regarding the wall time as a measure of anxiety-like behavior, anxiety was highest in the chronic mild stress condition, compared to the ST, CON, CS + Recovery and CS + ST conditions. By contrast, anxiety-like behavior was lowest in the ST condition, compared to the CS, CON, CS + Recovery and CS + ST conditions.

### 3.6. Number of Excrements

The number of excrements was significantly different between the five group conditions: (F (4, 25) = 11.38, *p* < 0.001) (Table 3). Post-hoc analyses showed that compared to the CON (0.83 ± 1.16), the number of excrements was statistically significantly longer in the CS (4.50 ± 1.22; *p* < 0.001) (Figure 9).

There was no significant difference between the control group (0.83 ± 1.16) and the CS + ST (0.83 ± 1.16; *p* = 1.000) and the ST (0.66 ± 0.81; *p* = 0.808) and the CS + R groups (2.00 ± 1.41; p=0.098), between the CS + ST group (0.83 ± 1.16) and the ST (0.66 ± 0.81; p=0.808) and CS + R groups (2.00 ± 1.41; *p* = 0.098), between the CS group (4.50 ± 1.22) and the CS + R group (2.00 ± 1.41; *p* = 0.061), between the ST group (0.66 ± 0.81) and the CS + R group (2.00 ± 1.41; *p* = 0.061).

To summarize, as regards the number of excrements as a parameter of anxiety, anxiety was highest in the CS condition, compared to the ST, CON, CS + Recovery, CS + ST conditions. By contrast, anxiety was lowest in the ST condition, compared to the CS, CON, CS + Recovery, CS + ST conditions.

### 3.7. Corticosterone

Corticosterone level significantly varied between the five group conditions: (F (4, 25) = 35.26, *p* < 0.001) (Table 3). Post-hoc analyses showed that: Compared to the CON (232.90 ± 20.66), corticosterone was statistically significantly greater in CS (323.55 ± 24.23; *p* < 0.001) and in the CS + ST (313.96 ± 34.13; *p* < 0.001). Compared to the CS + ST, corticosterone was significantly lower in the ST (208.26 ± 8.09; *p* < 0.001), and in the CS + Recovery condition (CS + Recovery (232.70 ± 28.60; *p* < 0.001) (Figure 10), so the effects evoked by CMS were at least partially recovered after a period without stress (i.e., CS + R group).

Compared to the CS, corticosterone was significantly lower in the ST (208.26 ± 8.09; *p* < 0.001) and in the CS + Recovery (232.70 ± 28.60; *p* < 0.001). While there was no significant difference between the ST condition and CON groups (232.90 ± 20.66; *p* = 0.097). Also, there were no significant differences between the CS + ST (313.96 ± 34.13) and CS groups (323.55 ± 24.23; *p* = 0.509) and also between the CS + ST (208.26 ± 8.09) and CS + Recovery groups (232.70 ± 28.60; *p* = 1.000).

To summarize, regarding corticosterone as a measure of anxiety-like behavior, anxiety was highest in the chronic mild stress condition, compared to the ST, CON, CS + Recovery and CS + ST conditions. By contrast, corticosterone was lowest in the ST condition, compared to the CS, CON, CS + Recovery and CS + ST groups.

## 4. Discussion

The purpose of present study was to evaluate the effect of chronic stress and possible preventive effect of swimming training on anxiety-related behaviors, indices and hormones, which was measured through an open field test, as well as the number of excrements and corticosterone levels. The findings of the study revealed that CMS (in the CS group) could increase anxiety-like behavior indices consisting of decrease of total distance, average speed and center time, and increase of corner time, number of excrements in the open field test and corticosterone levels. However, swimming training (in the ST group) might reduce anxiety-like behaviors including increase of the average speed, center time, and decrease of the corner time, wall time and number of excrements in open field test. Furthermore, swimming training could decrease the corticosterone levels compared to the CS + ST and CS conditions. Swimming training after a period of four weeks of CMS (in the CS + ST group) could decrease anxiety-like behaviors but these reductions were smaller than those of the ST group and were similar to the control group in some indices. Meanwhile, swimming training following CMS could increase corticosterone compared to CON, ST and CS + Recovery. A period of recovery time without any treatment after chronic stress (in the CS + Recovery group) could only reduce some anxiety-like behaviors compared to the CS condition and corticosterone compared to the CS and CS + ST groups.
One of the present study findings revealed the increasing effect of chronic stress on corticosterone. In CMS rats, swimming training not only could not reduce corticosterone, but also increased it compared to unstressed rats in ST and control groups and even compared to stressed rats in the CS + recovery group. There are some studies in humans and animals which confirm our findings considering the increasing effect of stress on cortisol or corticosterone.[2,3,4,5,6]

Elevated plasma cortisol levels are common in patients with anxiety [40]. In CMS models, the persistency of elevated plasma corticosterone levels has been demonstrated after stress withdrawal [41,42]. Elevated corticosterone in CS + ST, which was similar to the CS condition, can be justified by persistency of the effect of the two stressors, CMS and swimming training, on corticosterone, which continued for a relatively long time (eight weeks). Swimming training as exercise could increase corticosterone and this increase enhanced the increasing effect of CS on corticosterone and persisted for a long time.

In contrast, in unstressed rats, swimming training could reduce corticosterone. Some studies confirmed our findings and indicated the effect of exercise and physical activity on reducing anxiety [18,32] and cortisol and corticosterone [43,44]. It can be concluded that short durations of swimming training could suppress corticosterone and the possible increasing effect of four weeks of swimming on corticosterone was not persistent. 

The main mechanisms of physical and psychological stress-induced increase in corticosterone are related to the activity of the hypothalamic–pituitary–adrenal pathway. In this pathway, the hypothalamus stimulates the production of corticotropin-releasing factor (CRF), which in turn stimulates adrenocorticotropin hormone (ACTH) in the pituitary gland. ACTH stimulates the adrenal glands to release the cortisol hormone (corticosterone in rodents). Normally, when a stressor or a threat to organs is stopped, an integrated negative feedback loop system stops cortisol production, but exposure to long-term stressors causes continuous cortisol production and thus disorder. The negative feedback system in turn causes neural degeneration, inflammation, hippocampal volume reduction, anxiety and depression-like behaviors [17,18,28] and increased anxiety found in the present study.

However, regarding the reduction of corticosterone in the ST group of the present study, it can be concluded that exercise training can reduce resting cortisol. As in one study, the effect of exercise on cortisol concentrations in athletes and sedentary subjects was investigated and the results showed that cortisol concentration in athletes was lower than in sedentary individuals before exercise [45]. The decrease of corticosterone might be induced by modulating the activity of the hypothalamic–pituitary–adrenal pathway and negative feedback system or decreasing TNF-a levels, inflammation [46] and oxidative stress markers and increasing antioxidant activity of SOD, CAT, and glutathione peroxidase (GPx), neurotrophic expression [47]. In addition, moderate-intensity exercise can increase dopamine in the prefrontal cortex, which decreases corticosterone secretion [48].

The present study also showed that the swimming training group performed better in the open field test than the other groups, so that the traveled distance and the time spent at the center was better in ST than the other groups. Silva et al. (2019) confirmed our findings and indicated the effect of swimming training on reducing anxiety and also showed that swimming training improved mental health parameters (stress, depression), cognition, motor coordination and physical fitness [49]. However, two to four weeks of running in adult male mice increased center crossings and time spent in the center of the open field. Running training-induced changes in cognitive functioning are consistent with the benefits of exercise for anxiety-like behaviors [50,51]. Besides reduction of corticosterone, which was discussed in the previous section, another possible mechanism involved in this issue is endogenous opioids such as beta-endorphins. According to the endorphin hypothesis, continuous exercise with beta endorphin production from the hypothalamus and pituitary gland causes euphoria in the brain, which in turn reduces symptoms of anxiety [52].

Another interesting finding of the present study indicated that swimming training following chronic stress (CS + ST) could reduce some (but not all) anxiety-like behaviors compared to the control group. In the CS + ST group, corticosterone was not significantly different to the results for the CS group, which means that swimming training could probably reduce some but not all anxiety-like behaviors without changing corticosterone levels. Some other possible effects of swimming training on the brain resulted in psychological improvements (not related to corticosterone), including increases production of brain stem cells and their maturity in young rats [53,54], hippocampal neurogenesis [47], improved learning and memory (L&M) [55], increase of endorphins secretion from the brain [23], increased activity of the antioxidant enzyme [23], increased volume of the hippocampus [56], increased plasticity of the hippocampus and angiogenesis, increase in muscle myokines, liver hepatocytes and adipokines, butylated hydroxyanisole (BHA), β-hydroxybutyrate, brain-derived neurotrophic factor (BDNF), fibroblast growth factor-2 (FGF2), IGF-1, interleukin-6 (IL-6), interleukin-10 (IL-10) and vascular endothelial growth factor (VEGF) [57], as outcomes of exercise training are possible mediatory mechanisms of positive effect of swimming training on reducing anxiety indices which need to be clarified in future studies.

There are very many versions of the open field test [36] and their validity varies. However, the clear findings of this study justify the open field method that was used. However, a limitation of the present study was not measuring many possible physiological factors such as brain-derived neurotrophic factors (along with its receptor) in the hippocampus, as well as not measuring silent and astrogliosis neurons as parameters of brain function which are recommended to be measured in the future studies. 

## 5. Conclusions

According to open field findings, the present study shows the beneficial effects of swimming training on anxiety-like behaviors in rats under stress and rats not under stress, while the effect of swimming training was more prominent in unstressed rats. Swimming training decreased corticosterone in unstressed rats while increased it in stressed rats and the reason for this result needs to be clarified by future studies. Recovery time without any treatment compared to ST was not sufficient to reverse the effect of CMS anxiety behaviors, while reducing corticosterone. 

## Figures and Tables

**Figure 1 ijerph-17-06675-f001:**
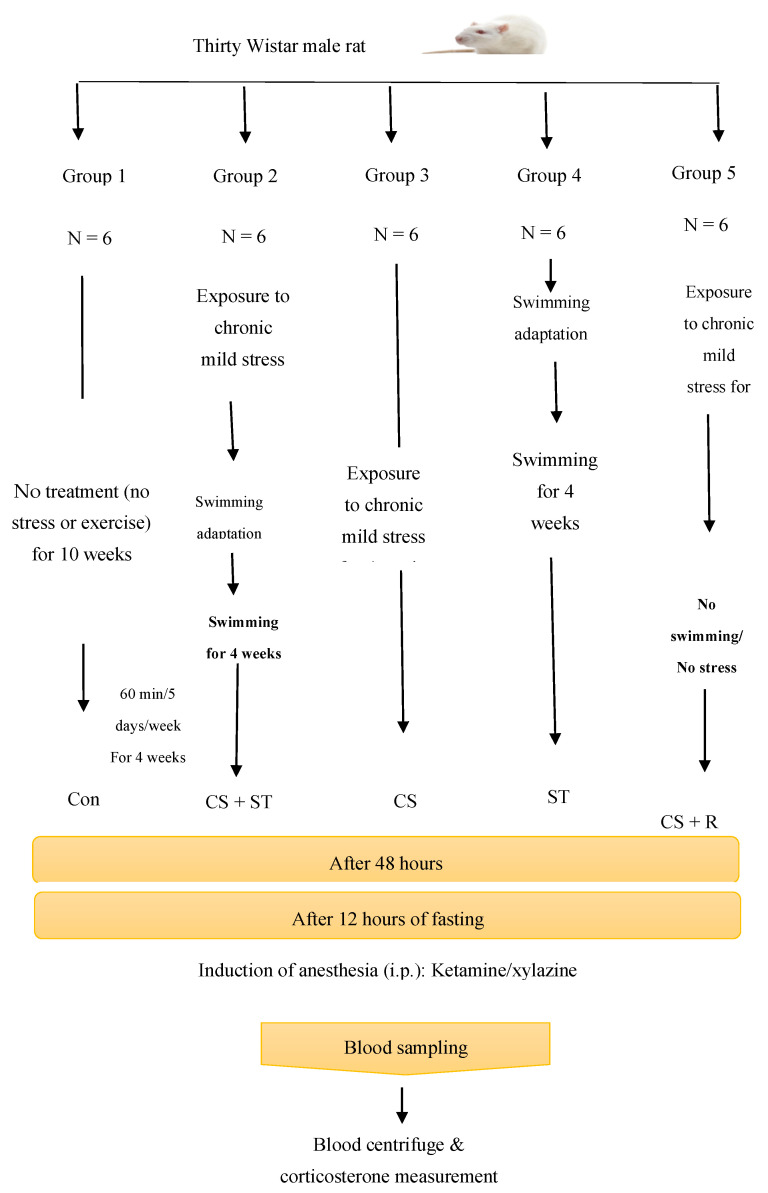
Schematic view of experimental procedure swimming training (ST); exposure to chronic mild stress (CS); exposure to chronic mild stress followed by swimming training (CS + ST); exposure to chronic mild stress followed by a recovery period (CS + R).

**Figure 2 ijerph-17-06675-f002:**
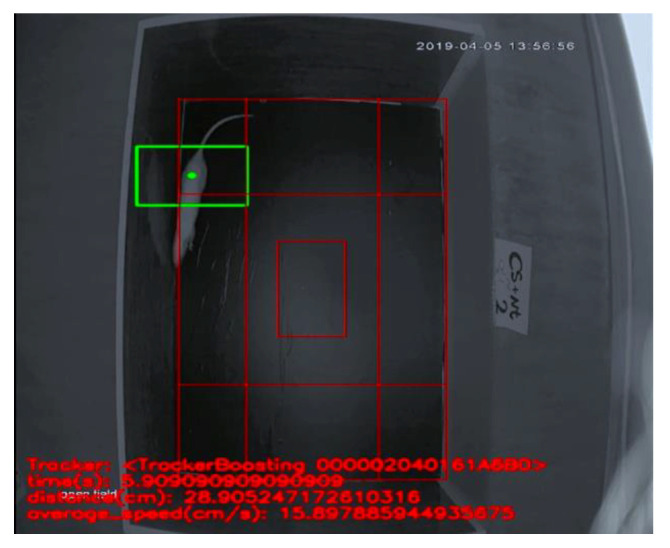
Open field test performance and box.

**Figure 3 ijerph-17-06675-f003:**
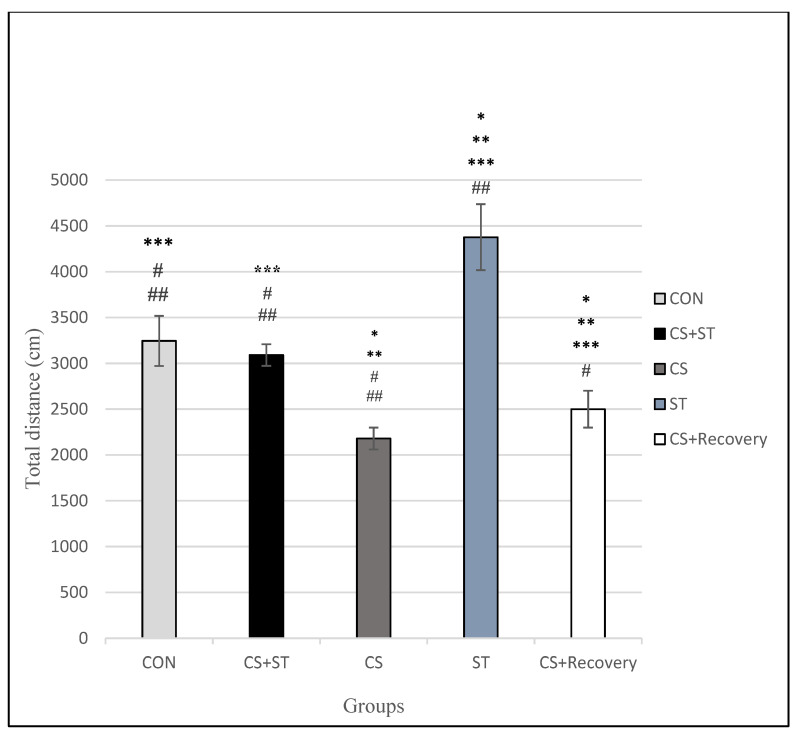
Total distance (cm) in male rats. Data are expressed as the mean ± S.E.M. * *p* < 0.00,1 significantly different from the control group. ** *p* < 0.001, significantly different from the CS + ST group. *** *p* < 0.001, significantly different from the CS group. **^#^**
*p* < 0.001, significantly different from the ST group. ^**##**^
*p* < 0.001, significantly different from the CS + Recovery group. CS + ST: chronic mild stress followed by swimming training, CS: chronic mild stress, ST: swimming training, CS + Recovery: chronic mild stress followed by recovery time, CON: control.

**Figure 4 ijerph-17-06675-f004:**
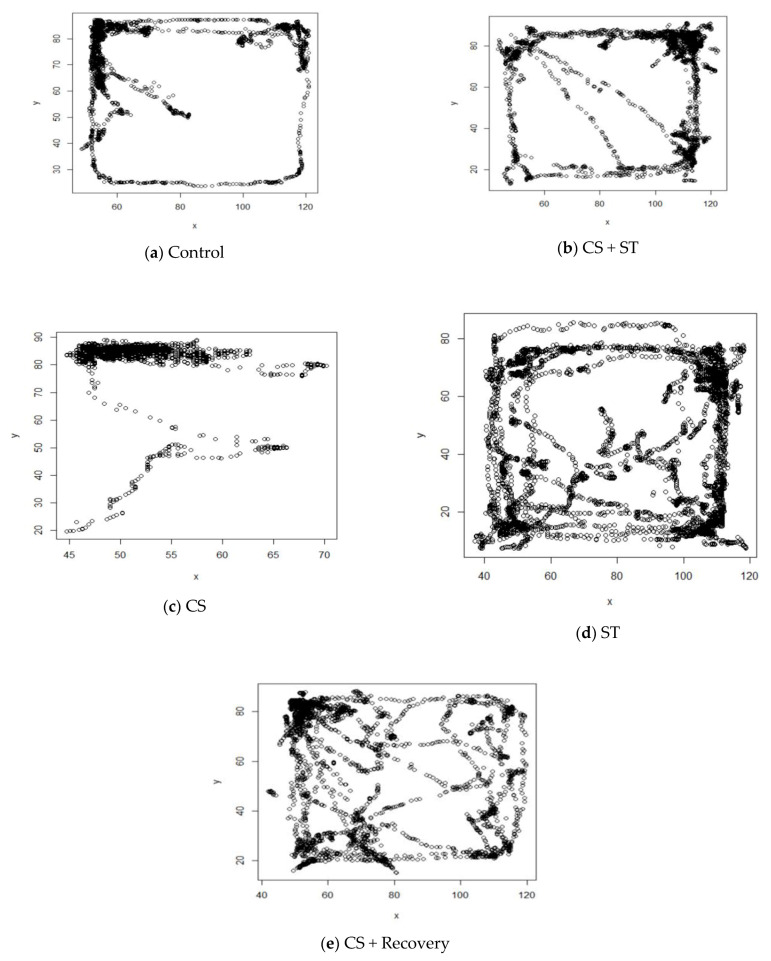
Schematic images of open field testing. (**a**) Control; (**b**) CS + ST: chronic mild stress followed by swimming training; (**c**) CS: chronic mild stress; (**d**) ST: swimming training; (**e**) CS + Recovery: chronic mild stress followed by recovery time, CON: control.

**Figure 5 ijerph-17-06675-f005:**
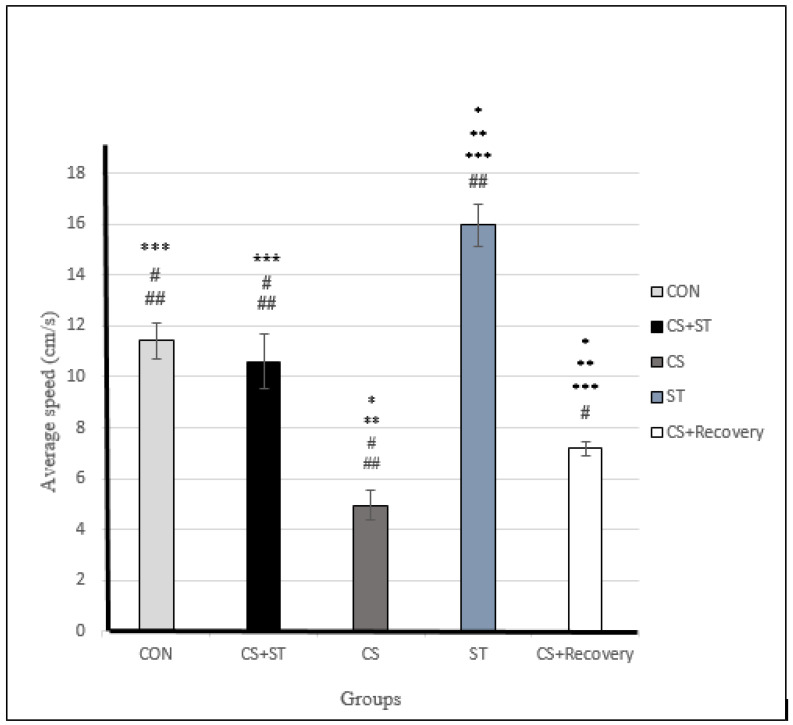
Average speed (cm/s) in male rats. Data are expressed as the mean ± S.E.M. * *p* < 0.001, significantly different from the control group. ** *p* < 0.001, significantly different from the CS + ST group. *** *p* < 0.001, significantly different from the CS group. **^#^**
*p* < 0.001, significantly different from the ST group. ^**##**^
*p* < 0.001, significantly different from the CS + Recovery group. CS + ST: chronic mild stress followed by swimming training, CS: chronic mild stress, ST: swimming training, CS + Recovery: chronic mild stress followed by recovery time. CON: control.

**Figure 6 ijerph-17-06675-f006:**
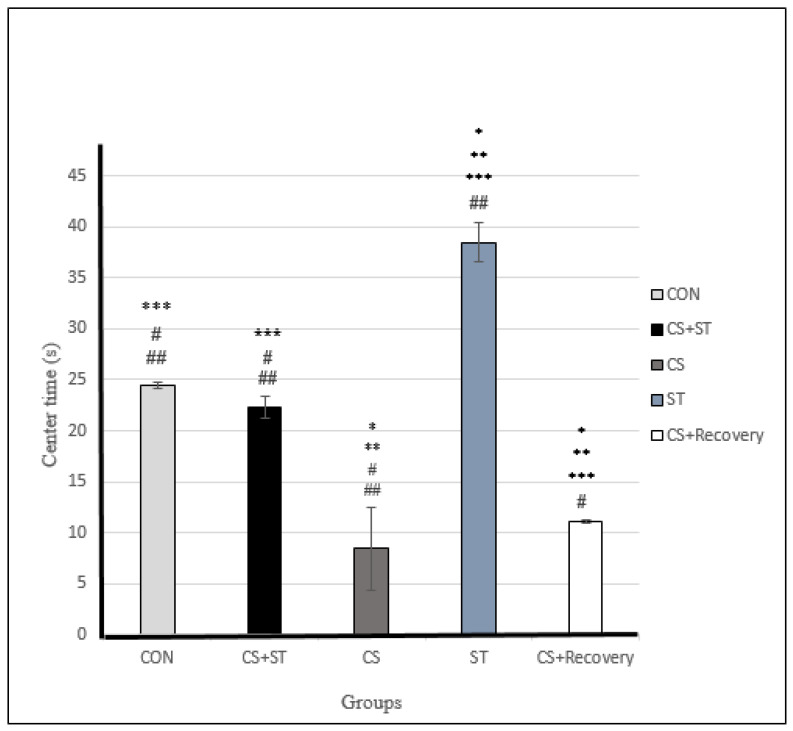
Center time (s) in male rats. Data are expressed as the mean ± S.E.M. * *p* < 0.001, significantly different from the control group. ** *p* < 0.001, significantly different from the CS + ST group. *** *p* < 0.001, significantly different from the CS group. **^#^**
*p* < 0.001, significantly different from the ST group. ^**##**^
*p* < 0.001, significantly different from the CS + Recovery group. CS + ST: chronic mild stress followed by swimming training, CS: chronic mild stress, ST: swimming training, CS + Recovery: chronic mild stress followed by recovery time, CON: control.

**Figure 7 ijerph-17-06675-f007:**
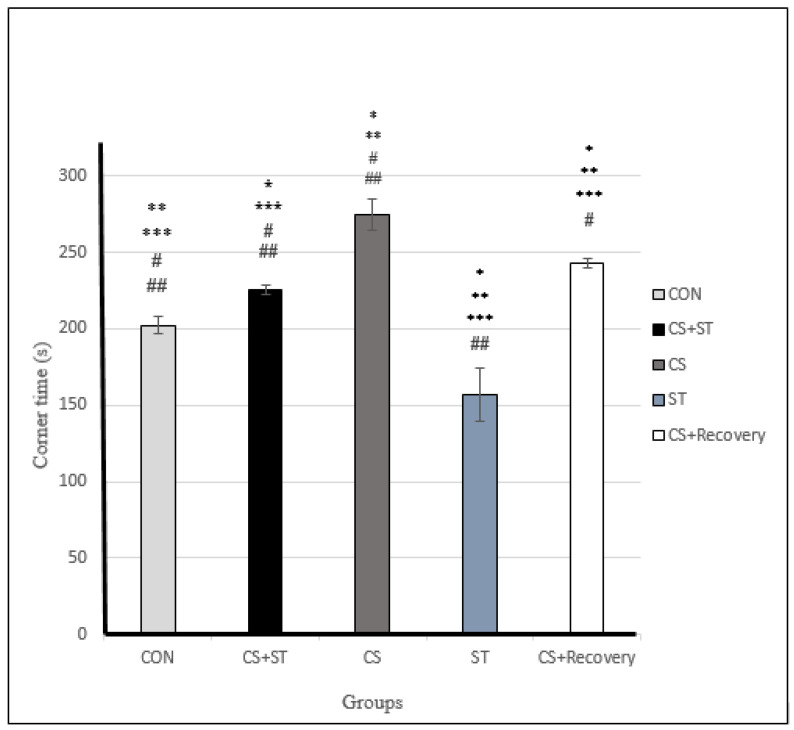
Corner time (s) in male rats. Data are expressed as the mean ± S.E.M. * *P* < 0.001 significantly different from the control group. ** *p* < 0.001 significantly different from the CS + ST group. *** *p* < 0.001, significantly different from the CS group. **^#^**
*p* < 0.001, significantly different from the ST group. *p* < 0.001, significantly different from the CS + Recovery group. CS + ST: chronic mild stress followed by swimming training, CS: chronic mild stress, ST: swimming training, CS + recovery: chronic mild stress followed by recovery time, CON: control.

**Figure 8 ijerph-17-06675-f008:**
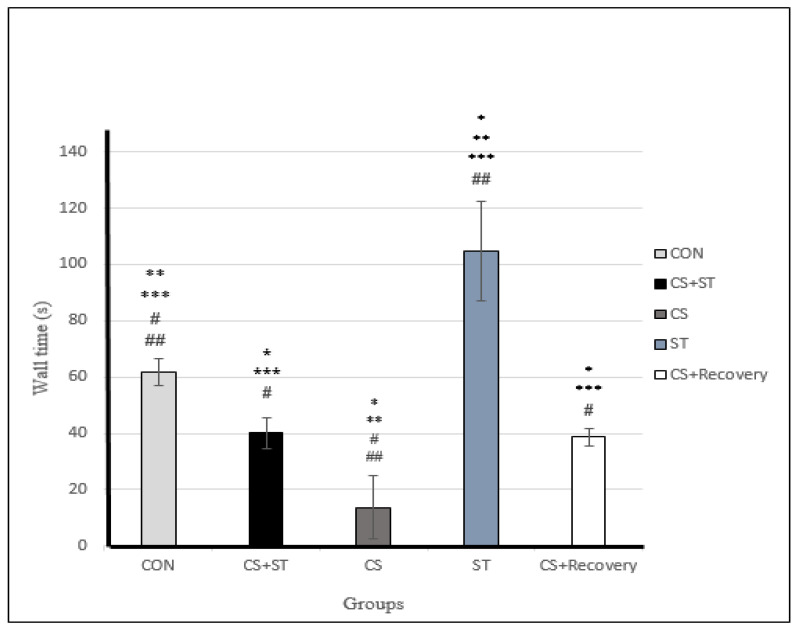
Wall time (s) in male rats. Data are expressed as the mean ± S.E.M. * *p* < 0.001, significantly different from the control group. ** *p* < 0.001, significantly different from the CS + ST group. *** *p* < 0.001, significantly different from the CS group. **^#^**
*p* < 0.001, significantly different from the ST group. ^**##**^
*p* < 0.001, significantly different from the CS + Recovery group. CS + ST: chronic mild stress followed by swimming training, CS: chronic mild stress, ST: swimming training, CS + Recovery: chronic mild stress followed by recovery time. CON: control.

**Figure 9 ijerph-17-06675-f009:**
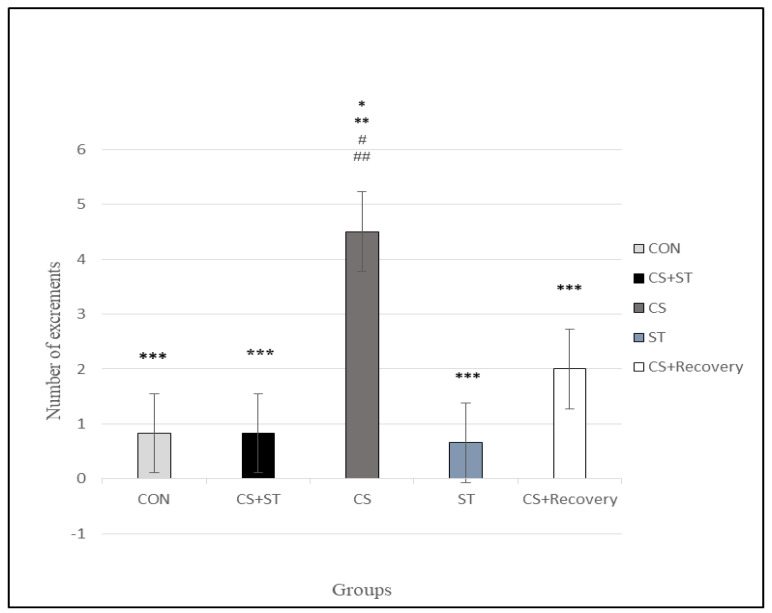
Number of excrements in male rats. Data are expressed as the mean ± S.E.M. * *p* < 0.001 significantly different from the control group. ** *p* < 0.001, significantly different from the CS + ST group. *** *p* < 0.001, significantly different from the CS group. **^#^**
*p* < 0.001, significantly different from the ST group. ^**##**^
*p* < 0.001, significantly different from the CS + Recovery group. CS + ST: exposed to chronic mild stress followed by swimming training, CS: chronic mild stress, ST: swimming training, CS + Recovery: exposed to chronic mild stress followed by recovery time. CON: control.

**Figure 10 ijerph-17-06675-f010:**
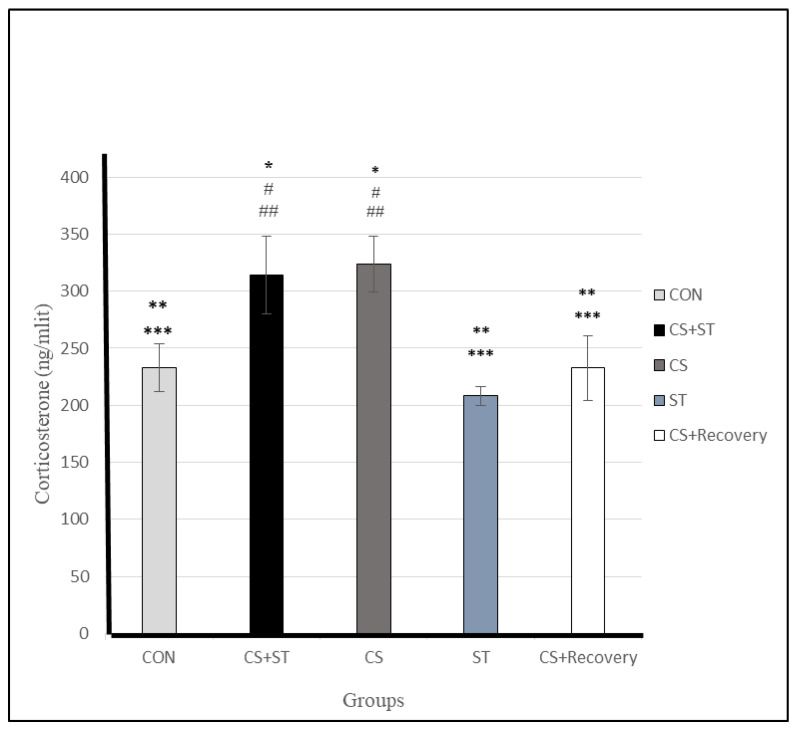
Total serum corticosterone (ng/mL) levels in male rats. Data are expressed as the mean ± S.E.M. * *p* < 0.001, significantly different from the control group. ** *p* < 0.001, significantly different from the CS + ST group. *** *p* < 0.001, significantly different from the CS group. **^#^**
*p* < 0.001, significantly different from the ST group. ^**##**^
*p* < 0.001, significantly different from the CS + Recovery group. CS + ST: chronic mild stress followed by swimming training, CS: chronic mild stress, ST: swimming training, CS + Recovery: chronic mild stress followed by recovery time, CON: control.

**Table 1 ijerph-17-06675-t001:** Water and swimming adaptation.

Adaptation to Water				Adaptation to the Swimming Training					
1st Day	2nd Day	3rd Day	4th Day	1st Day	2nd Day	3rd Day	4th Day	5th Day	6th Day
5 min swimming in shallow water	5 min swimming in head-height water	5 min swimming in deep water	15 min swimming in deep water	15 min of swimming exercise	24 min of swimming exercise	33 min of swimming exercise	42 min of swimming exercise	51 min of swimming exercise	60 min of swimming exercise

**Table 2 ijerph-17-06675-t002:** The chronic mild stress (CMS) protocol.

Day/Time	07:00	08:00	10:00	11:00	12:00	13:00	14:00	16:00	17:00	18:00
Sunday								Fd, 18 h		
Monday	Oi, 36 h		Fr, 15 h							
Tuesday					Ph, 2 h				Wet, 21 h	
Wednesday				Till. 1.5 h						Wd, 18 h
Thursday			Eb. 1.5 h							
Friday						Wd, 21 h				
Saturday		Eb. 1.5 h								

Abbreviations: Eb, exposure to empty bottle; Fd, food deprivation; Fr, food restriction; Oi, overnight illumination; Ph, paired housing; Til, tilted cage; Wd, water deprivation; Wet, wet cage.

**Table 3 ijerph-17-06675-t003:** ANOVA findings for the five groups of the study.

Parameter	Sum of Squares	Df	Mean Square	F	Sig.
Total distance	17138912.99	4	4284728.24	78.77	<0.001
Average speed	427.34	4	106.83	187.06	<0.001
Center time	3439.41	4	859.85	193.96	<0.001
Corner time	47013.02	4	11753.25	125.82	<0.001
Wall time	28033.22	4	7008.30	69.85	<0.001
Number of excrements	62.86	4	15.71	11.38	<0.001
Corticosterone	98386.06	4	24596.51	35.26	<0.001

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
