# Peer review of "The Effect of Swimming on Anxiety-Like Behaviors and Corticosterone in Stressed and Unstressed Rats"

_ijerph, 2020, doi:10.3390/ijerph17186675_

Round 1

Reviewer 1 Report

The research work titled “The effect of swimming on anxiety- like behaviors and corticosterone in stressed and unstressed rats” de Mohammad Amin and collaborators. Please make punctual and specific corrections where indicated: 2.1. Animals What was the electrical conductivity of the water (EC = dS/m) and what was the diet of the rats? 2.2. Swimming training protocol If the rats at the beginning of the experiment had a weight: 220 ± 20 g. Were rats weighed again during swimming days and time (adaptation)? I think the rats' effort at different swimming conditions could affect their weight. 2.4. Open field test Please make a graphic design of the experiment. 2.5. Blood sampling and corticosterone measurement What was the concentration of anesthesia (ketamine/xylazine) administered? The centrifuge is your property? Add models, make, and country of origin. 2.5. Blood sampling and corticosterone measurement Plasma corticosterone concentrations were determined by ELISA Kit… What were the plasma corticosterone concentrations used? Fig 2. The figure is very badly edited (This applies to all figures). References For references use: (software package, such as EndNote, Reference Manager or Zotero).

Author Response

Thank you for the thorough review, which helped us to improve the quality of the manuscript. Please find attached the detailed point-by-point-response. Thank you once again for all your kind efforts. 

Reviewer 2 Report

Interesting idea of research, although the methodological procedures are confused.

CMS procedure and CORT measurement raises reservations. If the authors chose the CMS procedure, they should additionally carry out a sucrose consumption test. The CMS model is used to induce anhedonia as measured by the amount of sucrose intake. Usually some animals do not respond to the CMS procedure – they are stress resilient. In the group of n = 6 it is difficult to conclude whether the actually applied procedure affected all rats. Wasn't it better to use restrain stress?
The procedure of the whole experiment is incomprehensible to me. One group of animals was adopted to swim prior to the experiment and then swam during 4 weeks, while the CS group of animals was trained after the CMS test and only swam 2 weeks. An after 48 h, OFT was then performed. After that, blood was taken for CORT.
Typically, the CORT is measurable immediately after stress and it's release is observed in minutes rather than days…The authors write: In CMS models, anxiety, depression, and elevated plasma corticosterone levels have been demonstrated typically at one, six, and 18 days after stress withdrawal [41,42]. The cited publications do not show this at all.

The error bars look the same and do not reflect the S.E.M. described in the text.
Statistics: A three-way ANOVA analysis should be performed considering the group of animals that swam and then recovery...

Author Response

(The authors gave the same response as above.)

Reviewer 3 Report

The design is good and the results are very clear, so publication is justified. A very large number of effects were obtained that support the notion that chronic stress occurred and affected corticosterone levels, but these effects were reversed by swimming training, and to some extent by a recovery period. The figures are easier to understand than the text in the Results section, and it might be better if the analyses of variance results were put into a table. Figure 7 needs a more complete caption.

At the beginning of the Discussion, a large number of specific hypotheses are stated.  These should have been included as part of a stronger rationale  at the end of the Introduction. The rest of the Discussion is rather disorganised and should be rewritten. It should be pointed out in the Discussion that there are very many versions of the open field test (see Ref 36) and their validity varies. However, the clear findings of this study justify the open field method that was used.

There are some errors of expression in the text, and some of these have been corrected in the attached version.

Author Response

(The authors gave the same response as above.)
